# Pattern of Distribution of Lymph Node Metastases in Individual Stations in Middle and Lower Gastric Carcinoma

**DOI:** 10.3390/cancers15072139

**Published:** 2023-04-04

**Authors:** Giuseppe Brisinda, Maria Michela Chiarello, Valeria Fico, Caterina Puccioni, Anna Crocco, Valentina Bianchi, Serafino Vanella

**Affiliations:** 1Dipartimento Scienze Mediche e Chirurgiche Addominali ed Endocrino Metaboliche, Fondazione Policlinico Universitario A. Gemelli, IRCCS, 00168 Roma, Italy; 2Dipartimento Universitario di Medicina e Chirurgia Traslazionale, Università Cattolica del Sacro Cuore, 00168 Roma, Italy; 3Unità Operativa di Chirurgia Generale, Dipartimento di Chirurgia, Azienda Sanitaria Provinciale, 87100 Cosenza, Italy; 4Unità Operativa di Chirurgia d’Urgenza e del Trauma, Dipartimento Scienze Mediche e Chirurgiche Addominali ed Endocrino Metaboliche, Fondazione Policlinico Universitario A. Gemelli, IRCCS, 00168 Roma, Italy; 5Unità Operativa di Chirurgia Oncologica della tiroide e della paratiroide, Istituto Nazionale Tumori, IRCCS Fondazione Pascale, 80131 Napoli, Italy; 6Unità Operativa di Chirurgia Generale e Oncologica, Azienda Ospedaliera San Giuseppe Moscati, 83100 Avellino, Italy

**Keywords:** gastric cancer, lymphadenectomy, lymph node stations, lymph node metastasis

## Abstract

**Simple Summary:**

Gastric cancer (GC) is a malignancy with great heterogenicity, and applying the same standard to all patients in different conditions may lead to treatment bias. D2 lymphadenectomy is the elective procedure for surgical treatment of GC. Middle- and lower-third GC show a different lymphatic spread than proximal tumors. This study has limitations. The retrospective nature of this study is a potential source of intrinsic biases. Our results must be interpreted with caution because they represent only a group of patients with distal GC. However, the present study shows that the primary tumor location is related to the location of nodal metastases. The knowledge of the localization of the tumor could lead to a tailored lymphadenectomy in the case of small tumors, in consideration of the absence of involvement of stations 7 to 12 in distal T1 cancers. On the other hand, in advanced GC, both circular and longitudinal localization should be considered to concentrate the lymphadenectomy on the most interesting individual stations and, in selected cases, extending the nodal dissection also to stations that are not routinely included in the D2 lymphadenectomy.

**Abstract:**

(1) Background: Lymph node (LN) dissection is the cornerstone of curative treatment of GC. The pattern of distribution of LN metastases is closely related to several factors. The aim of this study is to evaluate the factors determining the distribution of nodal metastases in a population of N+ distal GC patients undergoing gastrectomy and D2 lymphadenectomy. (2) Methods: The medical charts of 162 N+ GC patients who underwent surgical resection over a 15-year period were retrospectively analyzed. Clinical, pathological and anatomical characteristics were evaluated to identify the factors affecting the patterns and prevalence of metastases in individual LN stations. (3) Results: LN metastasis is correlated with the depth of the tumor and to diffuse-type tumors. A higher number of metastatic nodes was documented in patients with middle-third tumors (8.2 ± 7.3 vs. 4.5 ± 5.0 in lower-third tumors, *p* = 0.0001) and in patients with tumors located on the lesser curve. Station 4 showed the highest rate of metastases (53.1%). Concerning stations 7 to 12, station 8 showed the highest metastasis rate (28.4%). Metastases at stations 1, 2, 4 and 7 to 11 were dominant in middle-third cancer, whereas stations 5 and 6 were dominant in lower-third cancers. Station 4, 5, 6, 10 and 11 metastases were dominant when the cancer was located on the greater curve, whereas stations 1, 2, 7, 8 and 12 were dominant in lesser-curve cancers. (4) Conclusions: The study documented that in patients with distal GC, the distribution of nodal metastases at individual stations is closely related to primary tumor location.

## 1. Introduction

The presence of lymph node (LN) metastasis is one of the most significant prognostic factors in patients with gastric cancer (GC) [1,2,3]. Metastases occur during the early stages of the disease [3,4,5,6], and because their pre- and intra-operative diagnosis remain unreliable, a D2 lymphadenectomy should be performed whenever nodal involvement is suspected [6,7,8,9,10].

The classification of LN metastasis in patients with gastric cancer is controversial [9,10,11,12,13,14,15,16]. The numerical criterion has been introduced in place of the anatomical one, defining a minimum of 16 LNs retrieved as an adequate number in standard gastrectomy to ensure reliable N staging [17,18]. Even the minimum number of 16 retrieved nodes has been questioned, and there is not a widespread consensus [19,20,21]. Furthermore, guidelines define the extent of lymphadenectomy according to the type of gastrectomy regardless of the tumor location. It has been observed that the incidence of LN metastasis in GC is closely related to depth of invasion, tumor size and histopathological criteria [22,23]. Moreover, anatomical studies highlight a constant pattern of lymphatic flow in relation to the GC site [24,25]. Usually, proximal tumors metastasize predominantly to stations 1, 2, 7 and 9, while distal tumors metastasize to stations 5, 6 and 8 [16,26]. The distribution of locoregional LN metastases seems to be related to primary tumor location [25,27].

In this paper, the medical data of patients with histologically confirmed N+ distal GC, who underwent curative gastrectomy and D2 lymphadenectomy standardized according to the Japanese Gastric Cancer Association (JGCA) guidelines, were evaluated [18]. The primary endpoint of the study is to evaluate the factors determining the distribution and prevalence of nodal metastases. We investigated whether the circular and longitudinal location of the tumor could influence the distribution of metastases in the individual LN stations. Evaluations were also made to verify whether Lauren’s criteria, size of tumor and depth of invasion correlate with the distribution of LN metastases in a single station.

## 2. Materials and Methods

A retrospective analysis was conducted on GC patients who underwent gastrectomy between January 2008 and January 2022 at the General Surgery Operative Unit, Fondazione Policlinico Universitario “A Gemelli” IRCCS, and at the General Surgery Operative Unit, Azienda Sanitaria Provinciale Crotone. This study follows the STROBE reporting guidelines [28].

### 2.1. Inclusion Criteria

Eligible patients were included if they met the following inclusion criteria: 1. Primary distal GC; 2. radical gastrectomy and lymphadenectomy with curative intent (more than 16 LNs harvested); 3. the pathological report contained all essential information on the primary tumor and LN stations 1–12. All patients provided written consent before the surgical procedures. The study was conducted in accordance with the Declaration of Helsinki.

### 2.2. Exclusion Criteria

Patients with N- GC; with neoplasms other than gastric adenocarcinoma; with GC occurring in the esophagus–gastric junction or in the upper third of the gastric stump; those undergoing neoadjuvant treatment; or those with missing histopathological data were excluded.

### 2.3. Definition

The location of the tumor was defined according to the JGCA classification [29]. The histopathological classification followed the Lauren criteria [30]. The JGCA guidelines were used for the definition of LN stations and D2 lymphadenectomy [18]. The 8th Edition of the AJCC Cancer Staging system [17] was used for TNM classification.

### 2.4. Surgery and Histopathological Details

The gastrectomy procedure always included the removal of the greater and lesser omenta and perigastric LNs. Each LN station was removed, classified either during the operation or from the surgical specimen, collecting stations in separate containers (stations 1, 2 and 5 to 12) and markings on resection specimens (stations 3 and 4) and then submitted for histopathological examination. Stations 3a and 3b were grouped as station 3. Stations 4sa, 4sb and 4 d were grouped as station 4. Stations 11d and 11p were grouped as station 11. The number of retrieved and number of positive nodes per nodal station were carefully documented by the pathologist. The histological intestinal and mixed type were grouped as intestinal tumors. Histological type was categorized as differentiated or undifferentiated. Poorly differentiated tubular adenocarcinoma, signet ring cell adenocarcinoma and mucinous adenocarcinoma were considered to be undifferentiated. The T category was used to assess the depth of invasion. LN absence was defined as having no retrieved pathological LN despite precise surgical dissection without violation of guidelines, irrespective of metastasis.

### 2.5. Clinicopathological and Anatomical Data

The evaluated parameters included the patient age, sex, tumor size, tumor site at endoscopy, Lauren’s histological type, tumor differentiation, T and N stage, number of retrieved LNs and number of metastatic LNs. We identified subgroups in relation to age (≤65 years and >65 years) and tumor size (≤4 cm and >4 cm). We then grouped perigastric LNs (stations 1 to 6) into the Compartment I group and stations 7 to 12 into the Compartment II group. Moreover, we identified two subgroups with respect to longitudinal localization; specifically, the two subgroups identify tumors of the greater curvature (Gre) and tumors of the lesser curve (Less). We also identified two other subgroups in relation to the circular localization of the tumor, separating the tumors of the middle third (M—tumors in the distal two-thirds of the gastric corpus) from the tumors in the lower third (L—tumors in antrum or pylorus).

### 2.6. Aim of the Study

The primary endpoint of the study was to evaluate the factors determining the prevalence and distribution of nodal metastases. It was also investigated whether the position of the tumor in the identified subgroups influences the distribution and prevalence of LN metastases in each station. The evaluation of factors, such as histopathological criteria, size of the tumor and depth of invasion, was performed to identify a correlation with incidence of LN metastases and with distribution in individual station. It was also investigated whether the site of the primary tumor may have an influence on the presentation size of the tumor and whether the greater size of the tumor leads to a more extensive LN involvement.

### 2.7. Statistical Analysis

Data are expressed as a mean ± standard deviation (±SD) or as a percentage. Data were analyzed with standard statistical methods using GraphPad Prism Software (GraphPad, San Diego, CA, USA). Comparison of means ± SD was performed with the two-tailed *T*-test. A univariate analysis was performed on all potential factors influencing the course of the disease using the two-tailed Chi-square test or Fisher’s exact test for categorical data and the ANOVA test for continuous data in larger than two groups. A multivariate logistic regression was performed by constructing models that took into consideration the potential factors influencing the course of the disease that in the univariate analysis had a *p* value < 0.25, according to the Hosmer–Lemeshow rule. Furthermore, gender and age were included in the multivariate analysis because they are “confounders”. Regardless of the used test, a *p* value < 0.05 was considered statistically significant.

## 3. Results

Of the 201 patients who underwent gastrectomy for distal GC, 162 patients met our inclusion criteria. Reasons for exclusion were no complete pathological report of the LN metastases pattern attainable (16 cases) and no curative intent (23 cases).

Demographics, anatomical and pathological characteristics are shown in Table 1. The evaluated patients were 87 males and 75 females, aged 24 to 85 years old, with a mean age of 62.9 ± 12.9 years and no differences between the two sexes (males 63.9 ± 11.5 years vs. females 61.8 ± 14.3 years, *p* = 0.3). The average size of the tumor was 4.8 ± 2.1 cm (range: 1–12). The mean total nodal yield in stations 1–12 was 41.8 ± 8.3 (range 25–83) and the average number of metastatic LNs was 6.4 ± 6.5 (range 1–39). At Compartment I, an average of 24.1 ± 6.1 LNs (range 14–52) were removed. Of these, 4.2 ± 3.3 (range 1–19) were metastatic. At Compartment II, an average of 17.7 ± 4.1 (range 8–40) LNs were removed and, in 71 patients, 5.0 ± 4.4 LNs were found to be metastatic. Seventy-six patients (46.9%) underwent total gastrectomy and eighty-six underwent (53.1%) subtotal gastrectomy. No differences in range of tumor size (5.1 ± 2.3 cm in total gastrectomy and 4.5 ± 2.0 in subtotal gastrectomy, *p* = 0.06), number of retrieved LNs (42.0 ± 9.5 in total gastrectomy and 41.6 ± 7.2 in subtotal gastrectomy, *p* = 0.7), and number of metastatic LNs (7.3 ± 6.7 in total gastrectomy and 5.6 ± 6.3 in subtotal gastrectomy, *p* = 0.05) were found in these two groups of patients. In patients undergoing total gastrectomy, 24.2 ± 7.0 LNs of Compartment I were removed and in patients undergoing subtotal gastrectomy, 24.0 ± 5.3 LNs (*p* = 0.8). No differences were documented in the number of metastatic LNs at Compartment I level (4.5 ± 3.1 in total gastrectomy and 4.0 ± 3.4 in subtotal gastrectomy, *p* = 0.3). There were no differences between the two groups in the number of retrieved LNs of Compartment II (17.9 ± 4.7 in total gastrectomy and 17.6 ± 3.5 in subtotal gastrectomy, *p* = 0.6) and in the number of positive LNs (5.2 ± 4.8 in total gastrectomy and 4.7 ± 3.7 in subtotal gastrectomy, *p* = 0.6). The mean size of the tumor was greater in patients aged >65 years, patients with diffuse histotype, undifferentiated cancers, Gre tumors and M tumors. The size of the primary tumor increased in relation to the depth of the lesion, the T status, the N status and the AJCC/TNM stage (Table 2).

### 3.1. Total LNs

More LNs were excised in female patients, with no difference in the number of positive LNs with male patients. The number of retrieved LNs and number of metastatic LNs were directly related to the tumor size. The number of positive LNs was higher in diffuse forms (7.6 ± 7.1) than intestinal ones (5.1 ± 5.7, *p* = 0.008). The number of positive LNs was higher in patients with undifferentiated tumors (7.5 ± 7.2) than in patients with differentiated tumors (5.3 ± 5.5, *p* = 0.01—Table 3). In the two subgroups identified based on longitudinal localization, the number of positive LNs was higher in patients with Less cancers than in patients with Gre tumors. In relation to the circular location, the number of positive LNs was higher in M tumors compared to L tumors. The depth of the lesion, AJCC/TNM stage and N-status were directly related to the number of retrieved LNs. In the more advanced stages of the AJCC/TNM stage, more LNs were removed (IIIB 42.3 ± 5.9, IIIC 48.7 ± 9.4) than in the other stages (IB 39.3 ± 5.3, IIA 41.2 ± 6.0, IIB 41.6 ± 12.8, IIIA 41.0 ± 7.7, *p* = 0.02). T-status, the depth of the tumor and the AJCC/TNM stage were also directly related to the number of positive LNs (Table 3).

### 3.2. LNs at Compartments

Age did not show any influence on the number of retrieved LNs and on the number of positive nodes, both in the stations of Compartment I and in the stations of Compartment II (Table 4). More lymph nodes were removed in tumors >4 cm in Compartment I stations than in tumors ≤4 cm. The number of metastatic lymph nodes in each compartment was directly related to tumor size (Table 4). The number of positive LNs in Compartment I stations was higher in diffuse forms of cancer and in undifferentiated cancers. In the evaluation between the stations of Compartment I and the stations of Compartment II, the Gre or Less site of the primary tumor showed no differences. M tumors resulted in a higher number of positive LNs in Compartment I stations. N-status and AJCC/TNM stage were directly related to the number of retrieved LNs in Compartment I stations and Compartment II stations (Table 4). In tumors with serosa involvement, more LNs were removed than in tumors limited to the inner layers of the gastric wall. The number of retrieved LNs at Compartment I was directly related to T-status, depth, N-status and AJCC/TNM stage. These parameters were also directly related to the number of positive nodes for the stations of Compartment II (Table 4).

### 3.3. Individual Stations

LN involvement was limited to the perigastric stations in all T1 cancer. Age and gender had no influence on the distribution of metastases in each station. A larger tumor was documented in the case of stations 4, 5 and 6 metastasis (Table 5). In 62 patients (38.3%), LN involvement was limited to the perigastric stations. Of these stations, the most frequently affected were station 4 (86 patients) and station 3 (60 patients, Table 5). A difference was shown between diffuse and intestinal subtypes for stations 3 and 5. In relation to the circular location, M cancers involved stations 1 (10.8%), 2 (8.4%) and 4 (57.8%) compared to L cancers, which involved stations 3 (37.9%), 5 (24.0%) and 6 (8.8%). Considering the longitudinal location, stations 4 (78.3%), 5 (37.3%) and 6 (12.0%) were more frequently involved in Gre tumors compared to Less (Table 5).

The involvement of station 1 and 2 was observed only in advanced T3–4 tumors. Nine cases of LN absence were observed at station 1 (retrieved LNs 1.1 ± 0.5, metastatic LNs 1.3 ± 0.5). Five cases of LN absence were observed at station 2. Compared to M tumors, significantly less metastases were found for L tumors in stations 1 and 2 (Table 5). Twenty-one patients (37.5%) with station 3 involvement were found to be N1, while in the other patients metastasis at station 3 was associated with metastases in other stations. The involvement of station 4 was observed mainly in Gre tumors (65 cases). Station 5 was involved mainly in Gre tumors (Table 5). In Less cases, the tumor was N1. The Gre cases showed greater LN involvement (N2: 12 cases, 25%; N3a: 8 cases, 18.1%; N3b: 7 cases, 50%, *p* = 0.001). Station 6 metastasized in 10 Gre tumor patients (6.2%). Of these, seven were L tumors. The involvement of station 6 was always associated with involvement of other LN stations.

In stations 7 to 12, both diffuse histotype and undifferentiated tumors had no influence (Table 6). Tumor size correlates with metastases in stations 9, 10 and 11. The most frequently N+ were station 8 and station 12. M tumors metastasized most frequently to stations 7 (24.0% vs. 6.3% L tumor, *p* = 0.002), 8 (37.3% vs. 18.9%, *p* = 0.01), 9 (22.8% vs. 5.0%, *p* = 0.001), 10 (22.8% vs. 6.3%, *p* = 0.03) and 11 (18.0% vs. 2.5%, *p* = 0.001). Gre tumors metastasized predominantly to station 10 (26.5% vs. 2.5% Less tumor, *p* = 0.0001) and 11 (20.4% vs. 0%, *p* = 0.0001), while Less tumors to station 7 (22.7% vs. 8.4% Gre, *p* = 0.01), 8 (46.8% vs. 10.8%, *p* = 0.0001) and 12 (32.9% vs. 4.8%, *p* = 0.0001). The highest number of positive LNs were observed in case of involvement of station 9 (Table 6). These stations were found to be involved only in patients with N2, N3a or N3b advanced gastric cancers. The multivariate analysis showed that gender, age and size of the tumor have no influence on the distribution of positive nodes in the stations. The distribution of LN metastases in individual stations is influenced by both circular and longitudinal localization, depth of the tumor and T-stage (Table 7).

## 4. Discussion

D2 lymphadenectomy is considered the standard treatment for GC. A minimum of 16 LNs should be retrieved in order to accurately define the N stage. The number of retrieved LNs serves as a prognostic factor for GC, despite the fact that the optimal number of retrieved LNs remains controversial. Hayashi et al. [12] recommended the retrieval of more than 40 LNs after total gastrectomy for stage III patients, whereas Lu et al. [13] suggested that harvesting 21 LNs might represent a superior cut-off point for radical gastrectomy to better determine the prognosis of the patients. Total LN number is a powerful qualifier of staging and survival information for GC [31] without increased postoperative mortality [9].

The numeric-based N staging does not reflect the mapping of nodal dissection. An evaluation based exclusively on the number of LNs removed may be incomplete and, above all, compromise the therapeutic role of lymphadenectomy in cases of advanced GC. In evaluating the number of LNs in the stations grouped by compartments, we observed that the number of 16 LNs was easily reached even by removing only the perigastric stations. Furthermore, the true number of LNs that exist in each patient is variable and unknown. A mean of 33 LNs in D2 lymphadenectomy in GC-free individuals has previously been identified [32]. Due to reactive hyperplasia, even more LNs may be identifiable in GC patients. Studies of surgical dissection have shown that a mean of 26 LNs were removed with a D1 dissection, whereas 37.4 LNs were removed with a D2 lymphadenectomy [33]. Our results are consistent with these findings. In stations 1 to 12, we removed 41.8 ± 8.3 LNs, 24.1 ± 6.1 LNs from Compartment I and 17.7 ± 4.1 from Compartment II. Furthermore, the LNs number found at each station had high variation in nodal yields, and many stations contained no LNs despite adequate resection and pathological examination. Thus, it is difficult to define an ideal number of LNs in a surgical specimen. The type of LN dissection may significantly affect the number of LNs assessed [33], while the relationship between the tumor location and the number of LNs assessed was unclear.

Several studies investigated the pattern of LN metastasis in GC following curative lymphadenectomy. The results showed that development of LN metastasis is correlated with higher T-stage, diffuse-type tumors, the depth of invasion and tumor size. In the present study, an increasing number of LNs metastasis have been observed according to pathological depth of invasion (*p* = 0.001) and tumor size (*p* = 0.00004). An overall low prevalence of nodal metastases in T1 gastric carcinomas and an overall high prevalence in more advanced T3 and T4 carcinomas has also been documented. Furthermore, it has been found that the LN metastasis rate of undifferentiated tumors and of diffuse type are significantly higher as compared to that of high-differentiated tumors and of intestinal type.

Almost all previous studies have focused exclusively on the number of retrieved and metastatic LNs without proper consideration on their residing anatomic location groups [14,34]. However, the anatomical location of metastatic LNs is an important factor, especially in patients with stage N1-N2 disease [1]. Furthermore, it has been suggested that the required extent of LN dissection could vary per patient as the pattern of LN metastases may depend on tumor location [35,36]. The lymphatic flow is regular and directly related to the site of the tumor. In middle-third tumors located on the greater curve, the lymphatic flow is directed towards stations 4, 10 and 11, while in the case of middle-third tumors located on the lesser curve, the flow is directed towards stations 1 and 9. In lower-third tumors, two main pathways of lymphatic flow have been identified. Tumors located on the lesser curve involve stations 3, 5, 7 and 9, while in tumors located on the greater curve, involvement occurs at the level of the LN along the major and infrapyloric curvature and then those of the hepatic artery. We observed an involvement of station 5 predominantly in tumors located on the greater curve (37.3%), unlike what has been reported in the literature, where station 5 is mainly affected by tumors located on the lesser curve. These data could be linked to our sampling error or be related to the LN absence, which is reported in station 5 with a variable incidence up to over 60% [37,38,39,40]. The occurrence of a transversal metastasis, defined as metastasis to the opposite side of the tumor location without metastasis to the tumor side, is also likely. This event was significantly higher in lower third cancer (19/33 cases in present study) or when the tumor size was greater than 4 cm [41] (mean size 5.4 ± 2.2 cm in present study). As reported in Literature [16,25], we have documented that the LN involvement of each stations is correlated with the localization of the primary tumor, and both circular and longitudinal localization are directly related to the distribution of nodal metastases in multivariate analysis.

The station 6 is considered an important confluence of lymphatic channels that drain the distal two-thirds of the stomach [42]. Metastasis to station 6 is very common [43] and directly related to tumor size and location [44]. We observed metastases only in advanced GC of the greater curve. It has been demonstrated that station 4d and 6 metastases were independently associated with 14v metastasis [45]. In fact, the guideline emphasizes that D2 lymphadenectomy plus 14v dissection may be beneficial for patients who are suspected to harbor metastasis to station 6 [18]. Station 14v is anatomically downstream from 6 in the lymphatic flow for distal GC, and theoretically, once station 6 is invaded, 14v is at high risk of metastasis (incidence 17.1%) [46]. Although the impact on survival was not assessed in the present study, these findings suggest that station 14v should be dissected during gastrectomy for distal cancer with apparent metastasis to the infrapyloric nodes.

In the present study, the involvement of stations 7 to 12 was observed only in advanced GC. Metastases at “central LNs” (stations 8, 9 and 11) [47] was common (28.4% station 8, 14.2% station 9, 10.5% station 11) in our patients. The involvement was associated with larger tumor size (6.9 ± 2.1 cm in station 11), higher number of positive LNs (17.1 ± 8.5 LNs in patients with N+ station 9) and more advanced T stage. Moreover, the incidence of splenic hilar LNs metastasis in our patients was higher for middle-third cancer (22.8%, *p* = 0.003), tumors located on greater curve (26.5%, *p* = 0.0001) and increased tumor size (6.2 ± 2.3 cm), as reported in the literature [48,49]. Tumor diameter, histological type, T and N stages were also associated with 12a LN metastasis, with a reported rate ranged from 1.7% to 18.2% [50]. In the present study, N+ at station 12 was observed mainly in advanced GC located on the lesser curve.

## 5. Limitations

This study has limitations. Firstly, the sample size is small. Furthermore, the retrospective nature of this study is a potential source of intrinsic biases. Our results must be interpreted with caution because it represents only a group of patients with distal GC. We excluded from the evaluation cases with adenocarcinoma of the upper third of the stomach (Siewert III cancer). We are aware that it is commonly accepted that these tumors should be treated in the same way as GC (total gastrectomy with removal of the distal esophagus at least 5 cm above the macroscopic extent of the cancer) but were not included in the present study because the lymphatic flow in these is directed towards LN groups excluded from the standard D2 dissection. However, seeing the limited research regarding a more tailored approach in the treatment of distal GC, this study provides promising results and demonstrates the necessity for further research concerning the extent of lymphadenectomy. This preliminary work will need to be followed up with prospective and multicenter studies.

## 6. Conclusions

Our results show that the primary tumor location is related to the location of nodal metastases. GC is a malignancy with great heterogenicity, and applying the same standard to all patients in different conditions may lead to treatment bias. The knowledge of the localization of the tumor could lead to a tailored lymphadenectomy in the case of small tumors, in consideration of the absence of involvement of stations 7 to 12 in distal T1 cancers. On the other hand, in advanced GC, both circular and longitudinal localization should be considered to concentrate the lymphadenectomy on the most interesting individual stations and, in selected cases, extending the LN dissection also to stations that are not routinely included in the D2 lymphadenectomy.

## Figures and Tables

**Table 1 cancers-15-02139-t001:** Demographic and clinicopathological characteristics in 162 patients.

Parameters		Number of Patients	%
Age	≤65 years	86	53.1
	>65 years	76	46.9
Sex	Male	87	53.7
	Female	75	46.3
Tumor size	≤4 cm	73	45.1
	>4 cm	89	54.9
Lauren criteria	Diffuse	82	50.6
	Intestinal	80	49.4
Tumor grading	Differentiated	83	51.2
	Undifferentiated	79	48.8
Circular location	M	83	51.2
	L	79	48.8
Longitudinal location	Gre	83	51.2
	Less	79	48.8
T-status	T1	32	19.7
	T2	44	27.2
	T3	53	32.7
	T4	33	20.4
Depth	Mucosa	20	12.3
	Submucosa	12	7.4
	Muscolaris propria	44	27.2
	Subserosa	53	32.7
	Serosa	33	20.4
N-status	N1	56	34.5
	N2	48	29.6
	N3a	44	27.1
	N3b	14	8.8
AJCC/TNM stage	IB	32	19.7
	IIA	16	9.9
	IIB	29	17.9
	IIIA	30	18.5
	IIIB	41	25.3
	IIIC	14	8.7

All the patients were included in all evaluations. M: middle third. L: lower third. Gre: greater curvature. Less: lesser curvature.

**Table 2 cancers-15-02139-t002:** Tumor size in relation to the clinical, anatomical and pathological parameters considered.

Parameters		Size cm	*p*
Age	≤65 years	4.5 ± 1.9	0.04
	>65 years	5.1 ± 2.4	
Sex	Male	4.8 ± 2.2	0.6
	Female	4.9 ± 2.1	
Lauren criteria	Diffuse	5.1 ± 2.2	0.02
	Intestinal	4.5 ± 2.0	
Tumor grading	Differentiated	4.2 ± 1.8	0.0005
	Undifferentiated	5.4 ± 2.3	
Circular location	M	5.3 ± 2.2	0.002
	L	4.3 ± 2.0	
Longitudinal location	Gre	5.5 ± 2.4	0.00006
	Less	4.1 ± 1.6	
T-status	T1	3.3 ± 1.4	0.001
	T2	3.9 ± 1.1	
	T3	5.4 ± 1.9	
	T4	6.5 ± 2.5	
Depth	Mucosa	3.4 ± 1.6	0.0001
	Submucosa	3.0 ± 1.0	
	Muscolaris propria	3.9 ± 1.2	
	Subserosa	5.4 ± 1.9	
	Serosa	6.6 ± 2.5	
N-status	N1	4.4 ± 2.5	0.001
	N2	4.3 ± 1.9	
	N3a	5.3 ± 1.6	
	N3b	6.5 ± 2.4	
AJCC/TNM stage	IB	3.3 ± 1.5	0.02
	IIA	4.3 ± 1.0	
	IIB	4.3 ± 2.3	
	IIIA	5.6 ± 2.5	
	IIIB	5.4 ± 1.6	
	IIIC	6.5 ± 2.4	

Values are mean ± SD. All the patients were included in all evaluations. M: middle third. L: lower third. Gre: greater curvature. Less: lesser curvature.

**Table 3 cancers-15-02139-t003:** Number of retrieved LNs and number of metastatic LNs, in relation to the clinical, anatomical and pathological parameters considered.

		Retrieved LNs	*p*	Metastatic LNs	*p*
Age	≤65 years	42.2 ± 9.5	0.2	6.6 ± 7.2	0.3
	>65 years	41.3 ± 6.7		6.1 ± 5.7	
Sex	Male	40.4 ± 5.7	0.01	6.0 ± 5.0	0.4
	Female	43.5 ± 10.4		6.9 ± 8.0	
Tumor size	≤4 cm	39.9 ± 5.8	0.01	4.1 ± 4.0	0.00004
	>4 cm	43.3 ± 9.7		8.2 ± 7.5	
Lauren criteria	Diffuse	42.3 ± 10.1	0.1	7.6 ± 7.1	0.008
	Intestinal	41.2 ± 5.9		5.1 ± 5.7	
Tumor grading	Differentiated	42.1 ± 9.2	0.2	5.3 ± 5.5	0.01
	Undifferentiated	41.4 ± 7.4		7.5 ± 7.2	
Circular location	M	42.6 ± 9.8	0.2	8.2 ± 7.3	0.0001
	L	41.0 ± 6.3		4.5 ± 5.0	
Longitudinal location	Gre	41.7 ± 9.7	0.9	5.8 ± 6.3	0.02
	Less	41.8 ± 6.6		7.0 ± 6.8	
T-status	T1	39.3 ± 5.3	0.1	1.5 ± 0.5	0.001
	T2	41.3 ± 11.0		3.7 ± 2.9	
	T3	42.1 ± 8.0		8.7 ± 6.3	
	T4	44.2 ± 6.2		11.0 ± 8.6	
Depth	Mucosa	38.5 ± 3.9	0.001	1.3 ± 0.5	0.001
	Submucosa	40.7 ± 7.1		1.6 ± 0.5	
	Muscolaris propria	41.3 ± 11.0		3.7 ± 2.9	
	Subserosa	42.1 ± 8.0		8.7 ± 6.3	
	Serosa	44.2 ± 6.2		11.0 ± 8.6	
N-status	N1	40.1 ± 5.5	0.006	1.5 ± 0.5	0.0001
	N2	41.8 ± 11.2		4.1 ± 1.0	
	N3a	41.7 ± 6.0		9.8 ± 2.4	
	N3b	48.7 ± 9.4		22.8 ± 7.2	
AJCC/TNM stage	IB	39.3 ± 5.3	0.02	1.4 ± 0.5	0.001
	IIA	41.2 ± 6.0		1.5 ± 0.5	
	IIB	41.6 ± 12.8		3.7 ± 1.3	
	IIIA	41.0 ± 7.7		4.6 ± 3.0	
	IIIB	42.3 ± 5.9		9.7 ± 2.3	
	IIIC	48.7 ± 9.4		22.8 ± 7.2	

Values are mean ± SD. All the patients were included in all evaluations. LN: lymph node. M: middle third. L: lower third. Gre: greater curvature. Less: lesser curvature.

**Table 4 cancers-15-02139-t004:** Number of retrieved LNs and metastatic LNs in compartment I (stations 1 to 6) and compartment II (stations 7 to 12), in relation to in relation to the clinical, anatomical and pathological parameters considered.

Parameters		Compartment I				Compartment II			
		Retrieved LNs	*p*	Metastatic LNs	*p*	Retrieved LNs	*p*	Metastatic LNs	*p*
Age	≤65 years	24.1 ± 7.0	0.4	4.3 ± 3.5	0.5	18.1 ± 4.7	0.2	5.4 ± 5.2	0.4
	>65 years	24.0 ± 5.1		4.0 ± 3.0		17.3 ± 3.3		4.6 ± 3.1	
Sex	Male	23.5 ± 5.0	0.08	3.9 ± 2.7	0.2	16.9 ± 3.0	0.008	4.4 ± 2.5	0.2
	Female	24.8 ± 7.2		4.5 ± 3.8		18.6 ± 4.9		5.8 ± 6.0	
Tumor size	≤4 cm	22.6 ± 4.7	0.005	3.1 ± 2.3	0.00005	17.3 ± 3.3	0.2	3.6 ± 2.4	0.04
	>4 cm	25.3 ± 6.9		5.1 ± 3.6		18.0 ± 4.6		5.5 ± 4.9	
Lauren	Diffuse	24.7 ± 7.3	0.1	4.9 ± 3.8	0.007	17.6 ± 4.3	0.8	5.1 ± 3.7	0.7
	Intestinal	23.5 ± 4.7		3.5 ± 2.4		17.8 ± 3.9		4.8 ± 5.3	
Tumor grading	Differentiated	24.1 ± 6.8	0.9	3.5 ± 2.4	0.008	18.0 ± 4.4	0.3	4.6 ± 5.0	0.4
	Undifferentiated	24.1 ± 5.3		4.8 ± 3.9		17.3 ± 3.7		5.3 ± 3.8	
Circular location	M	24.7 ± 7.1	0.1	5.0 ± 3.5	0.0007	17.9 ± 4.8	0.5	5.2 ± 4.6	0.2
	L	23.4 ± 4.8		3.3 ± 2.6		17.5 ± 3.2		4.4 ± 3.7	
Longitudinal location	Gre	24.1 ± 7.0	0.9	4.0 ± 3.4	0.5	17.6 ± 4.3	0.7	5.0 ± 3.8	0.9
	Less	24.0 ± 5.0		4.4 ± 3.1		17.8 ± 3.9		5.0 ± 4.8	
T-status	T1	21.7 ± 4.1	0.03	1.5 ± 0.5	0.001	17.5 ± 3.0	0.7	-	0.04
	T2	24.1 ± 7.8		3.2 ± 2.0		17.3 ± 4.7		2.7 ± 1.9	
	T3	24.3 ± 5.9		5.5 ± 3.4		17.9 ± 3.7		4.6 ± 3.0	
	T4	26.1 ± 5.1		6.2 ± 3.6		18.3 ± 4.8		6.4 ± 6.0	
Depth	Mucosa	21.1 ± 3.2	0.05	1.3 ± 0.5	0.00001	17.3 ± 3.0	0.8	-	0.04
	Submucosa	22.7 ± 5.4		1.7 ± 0.5		17.9 ± 3.0		-	
	Muscolaris propria	24.1 ± 7.8		3.1 ± 2.0		17.3 ± 4.7		2.7 ± 1.9	
	Subserosa	24.3 ± 5.9		5.4 ± 3.4		17.9 ± 3.7		4.6 ± 3.0	
	Serosa	28.3 ± 6.7		6.2 ± 3.6		18.3 ± 4.8		6.4 ± 6.0	
N-status	N1	22.6 ± 4.6	0.01	1.5 ± 0.5	0.001	17.4 ± 3.1	0.04	-	0.001
	N2	24.8 ± 7.8		3.6 ± 0.9		17.1 ± 4.6		1.6 ± 0.6	
	N3a	23.9 ± 5.0		5.9 ± 2.0		17.8 ± 2.8		4.0 ± 1.4	
	N3b	28.3 ± 6.7		11.3 ± 4.0		20.5 ± 7.3		11.5 ± 5.9	
AJCC/TNM stage	IB	21.7 ± 4.1	0.03	1.5 ± 0.5	0.001	17.5 ± 3.0	0.04	-	0.001
	IIA	23.9 ± 5.2		1.5 ± 0.5		17.3 ± 3.5		-	
	IIB	24.0 ± 8.8		3.4 ± 1.3		17.6 ± 5.1		1.6 ± 0.5	
	IIIA	24.8 ± 5.9		3.8 ± 1.9		16.3 ± 3.4		2.4 ± 1.9	
	IIIB	24.2 ± 4.9		5.8 ± 1.9		18.2 ± 2.6		3.9 ± 1.3	
	IIIC	28.3 ± 6.7		11.3 ± 4.0		20.5 ± 7.3		11.5 ± 6.0	

Values are mean ± SD. All the patients were included in all evaluations. LN: lymph node. M: middle third. L: lower third. Gre: greater curvature. Less: lesser curvature.

**Table 5 cancers-15-02139-t005:** Incidence of LN metastasis per perigastric nodal station.

Parameters	Overall	Station 1	Station 2	Station 3	Station 4	Station 5	Station 6
Number of patients (%)	162 (100%)	9 (5.5%)	8 (4.9%)	60 (37.0%)	86 (53.1%)	33 (20.4%)	10 (6.2%)
Age year	62.9 ± 12.9	63.5 ± 10.0	62.6 ± 18.7	62.5 ± 11.5	63.9 ± 12.2	59.6 ± 16.6	64.0 ± 15.7
		*p* = 0.8	*p* = 0.9	*p* = 0.7	*p* = 0.3	*p* = 0.3	*p* = 0.7
Sex M/F	87M/75F	7M/2F	4M/4F	32M/28F	48M/38F	20M/13F	4M/6F
		*p* = 0.1	*p* = 0.5	*p* = 0.5	*p* = 0.7	*p* = 0.5	*p* = 0.2
Tumor size cm	4.8 ± 2.1	4.9 ± 0.8	5.5 ± 1.2	4.1 ± 1.6	5.3 ± 2.4	5.4 ± 2.2	6.8 ± 2.4
		*p* = 0.3	*p* = 0.3	*p* = 0.008	*p* = 0.007	*p* = 0.08	*p* = 0.002
Lauren							
Diffuse	82	3 (3.6%)	2 (2.4%)	23 (28.0%)	49 (59.7%)	22 (26.8%)	8 (9.7%)
Intestinal	80	6 (7.5%)	6 (7.5%)	37 (46.2%)	37 (46.2%)	11 (13.7%)	2 (2.5%)
		*p* = 0.2	*p* = 0.1	*p* = 0.01	*p* = 0.08	*p* = 0.03	*p* = 0.05
Tumor grading							
Differentiated	83	3 (3.6%)	3 (3.6%)	33 (39.7%)	43 (51.8%)	14 (16.8%)	4 (4.8%)
Undifferentiated	79	6 (7.5%)	5 (6.3%)	27 (34.1%)	43 (54.4%)	19 (24.0%)	6 (7.6%)
		*p* = 0.2	*p* = 0.4	*p* = 0.5	*p* = 0.7	*p* = 0.3	*p* = 0.5
Circular location							
M	83	9 (10.8%)	7 (8.4%)	30 (36.1%)	48 (57.8%)	14 (16.8%)	3 (3.6%)
L	79	0	1 (1.2%)	30 (37.9%)	38 (48.1%)	19 (24.0%)	7 (8.8%)
		*p* = 0.001	*p* = 0.03	*p* = 0.8	*p* = 0.2	*p* = 0.3	*p* = 0.2
Longitudinal location							
Gre	83	0	3 (3.6%)	30 (36.1%)	65 (78.3%)	31 (37.3%)	10 (12.0%)
Less	79	9 (11.4%)	5 (6.3%)	30 (37.9%)	21 (26.5%)	2 (2.5%)	0
		*p* = 0.001	*p* = 0.4	*p* = 0.8	*p* = 0.0001	*p* = 0.0001	*p* = 0.001
Total LNs							
Retrieved	41.8 ± 8.3	44.8 ± 3.2	47.3 ± 6.9	42.8 ± 7.3	41.8 ± 8.1	42.0 ± 11.5	45.2 ± 9.6
		*p* = 0.2	*p* = 0.06	*p* = 0.2	*p* = 0.9	*p* = 0.4	*p* = 0.1
Metastatic	6.4 ± 6.5	13.0 ± 4.9	14.5 ± 10.7	6.8 ± 7.4	6.8 ± 6.7	7.6 ± 8.0	13.0 ± 9.5
		*p* = 0.003	*p* = 0.001	*p* = 0.7	*p* = 0.6	*p* = 0.3	*p* = 0.003
LNs at each station							
Retrieved	NA	1.1 ± 0.5	1.3 ± 0.5	8.0 ± 2.5	9.7 ± 3.1	2.4 ± 1.9	2.0 ± 1.7
Metastatic	NA	1.3 ± 0.5	1.1 ± 0.3	3.9 ± 2.5	3.8 ± 2.7	2.4 ± 1.9	2.0 ± 1.7
T status							
T1	32	0	0	15 (46.8%)	15 (46.8%)	3 (9.3%)	0
T2	44	0	0	13 (29.5%)	22 (50.0%)	12 (27.2%)	2 (4.5%)
T3	53	3 (5.6%)	4 (7.5%)	18 (33.9%)	31 (58.4%)	14 (26.4%)	7 (13.2%)
T4	33	6 (18.1%)	4 (12.1%)	14 (42.4%)	18 (54.5%)	4 (12.1%)	1 (3.0%)
		*p* = 0.001	*p* = 0.006	*p* = 0.8	*p* = 0.3	*p* = 0.8	*p* = 0.2
Depth		.					
Mucosa	20	0	0	10 (50.0%)	9 (45.0%)	1 (5.0%)	0
Submucosa	12	0	0	5 (41.6%)	6 (50.0%)	2 (16.6%)	0
Muscolaris propria	44	0	0	13 (29.5%)	22 (50.0%)	12 (27.2%)	2 (4.5%)
Subserosa	53	3 (5.6%)	4 (7.5%)	18 (33.9%)	31 (58.4%)	14 (26.4%)	7 (13.2%)
Serosa	33	6 (18.1%)	4 (12.1%)	14 (42.4%)	18 (54.5%)	4 (12.1%)	1 (3.0%)
		*p* = 0.001	*p* = 0.005	*p* = 0.09	*p* = 0.09	*p* = 0.9	*p* = 0.7
N status							
N1	56	0	0	21 (37.5%)	30 (53.7%)	6 (10.7%)	0
N2	48	0	0	13 (27.0%)	27 (56.2%)	12 (25.0%)	3 (6.2%)
N3a	44	6 (13.6%)	6 (14.6%)	21 (47.7%)	22 (50.0%)	8 (18.1%)	4 (9.0%)
N3b	14	3 (21.4%)	2 (14.2%)	5 (35.7%)	7 (50.0%)	7 (50.0%)	3 (21.4%)
		*p* = 0.001	*p* = 0.001	*p* = 0.5	*p* = 0.6	*p* = 0.01	*p* = 0.002
AJCC/TNM stage							
IB	32	0	0	15 (46.8%)	15 (46.8%)	3 (9.3%)	0
IIA	16	0	0	6 (37.5%)	7 (43.7%)	3 (18.7%)	0
IIB	29	0	0	7 (24.1%)	16 (55.1%)	8 (27.5%)	1 (3.4%)
IIIA	30	0	0	6 (20.0%)	22 (73.3%)	5 (16.6%)	3 (10.0%)
IIIB	41	6 (14.6%)	6 (14.6%)	21 (51.2%)	19 (46.3%)	7 (17.0%)	3 (7.3%)
IIIC	14	3 (21.4%)	2 (14.2%)	5 (35.7%)	7 (50.0%)	7 (50.0%)	3 (21.4%)
		*p* = 0.001	*p* = 0.001	*p* = 0.5	*p* = 0.7	*p* = 0.4	*p* = 0.04

Values are mean ± SD. *p* values: comparison with values in overall patients. All the patients were included in all evaluations. Numbers in brackets represents percentage. LN: lymph node. M: mi dle-third. L: lower third. Gre: greater curvature. Less: lesser curvature. NA: not applicable.

**Table 6 cancers-15-02139-t006:** Incidence of LNs metastasis per nodal station 7 to 12.

	Overall	Station 7	Station 8	Station 9	Station 10	Station 11	Station 12
Number of patients (%)	162 (100%)	25 (15.4%)	46 (28.4%)	23 (14.2%)	24 (14.8%)	17 (10.5%)	30 (18.5%)
Age year	62.9 ± 12.9	66.5 ± 8.8	64.3 ± 19.9	64.4 ± 11.6	61.3 ± 15.3	62.5 ± 15.2	62.1 ± 11.4
		*p* = 0.2	*p* = 0.4	*p* = 0.5	*p* = 0.4	*p* = 0.8	*p* = 0.7
Sex M/F	87M/75F	13M/12F	27M/19F	16M/7F	12M/12F	10M/7F	20M/10F
		*p* = 0.8	*p* = 0.4	*p* = 0.09	*p* = 0.7	*p* = 0.6	*p* = 0.1
Tumor size cm	4.8 ± 2.1	5.3 ± 1.9	5.3 ± 1.9	5.8 ± 2.3	6.2 ± 2.3	6.9 ± 2.1	5.0 ± 1.5
		*p* = 0.2	*p* = 0.1	*p* = 0.002	*p* = 0.001	*p* = 0.001	*p* = 0.1
Lauren							
Diffuse	82	13 (15.8%)	26 (31.7%)	14 (17.1%)	17 (20.7%)	13 (15.8%)	18 (21.9%)
Intestinal	80	12 (15.0%)	20 (25.0%)	9 (11.2%)	7 (8.7%)	4 (5.0%)	12 (15.0%)
		*p* = 0.8	*p* = 0.3	*p* = 0.3	*p* = 0.03	*p* = 0.02	*p* = 0.2
Tumor grading							
Differentiated	83	10 (12.0%)	19 (22.8%)	9 (10.8%)	10 (12.0%)	7 (8.4%)	14 (16.8%)
Undifferentiated	79	15 (18.9%)	27 (34.1%)	14 (17.7%)	14 (17.7%)	10 (12.6%)	16 (20.2%)
		*p* = 0.2	*p* = 0.1	*p* = 0.2	*p* = 0.3	*p* = 0.3	*p* = 0.6
Circular location							
M	83	20 (24.0%)	31 (37.3%)	19 (22.8%)	19 (22.8%)	15 (18.0%)	18 (21.7%)
L	79	5 (6.3%)	15 (18.9%)	4 (5.0%)	5 (6.3%)	2 (2.5%)	12 (15.1%)
		*p* = 0.002	*p* = 0.009	*p* = 0.001	*p* = 0.002	*p* = 0.001	*p* = 0.3
Longitudinal location							
Gre	83	7 (8.4%)	9 (10.8%)	9 (10.8%)	22 (26.5%)	17 (20.4%)	4 (4.8%)
Less	79	18 (22.7%)	37 (46.8%)	14 (17.7%)	2 (2.5%)	0	26 (32.9%)
		*p* = 0.01	*p* = 0.0001	*p* = 0.2	*p* = 0.0001	*p* = 0.0001	*p* = 0.0001
Total LNs							
Retrieved	41.8 ± 8.3	46.2 ± 8.1	45.0 ± 7.6	44.1 ± 9.2	44.0 ± 12.3	44.8 ± 9.6	45.0 ± 7.5
		*p* = 0.01	*p* = 0.01	*p* = 0.2	*p* = 0.2	*p* = 0.1	*p* = 0.04
Metastatic	6.4 ± 6.5	15.6 ± 8.6	13.0 ± 7.8	17.1 ± 8.5	12.3 ± 8.0	13.8 ± 7.6	14.1 ± 7.9
		*p* = 0.0001	*p* = 0.0001	*p* = 0.0001	*p* = 0.0001	*p* = 0.0001	*p* = 0.0001
LNs at each station							
Retrieved	NA	2.7 ± 1.0	2.5 ± 1.1	3.1 ± 0.9	3.2 ± 0.9	3.0 ± 1.1	3.0 ± 1.1
Metastatic	NA	2.2 ± 1.1	2.0 ± 1.5	2.5 ± 1.4	2.2 ± 0.9	1.8 ± 1.2	2.2 ± 1.4
T status							
T1	32	0	0	0	0	0	0
T2	44	2 (4.5%)	5 (11.3%)	2 (4.5%)	2 (4.5%)	0	2 (4.5%)
T3	53	11 (20.7%)	23 (43.3%)	10 (18.8%)	15 (28.3%)	11 (20.7%)	16 (30.1%)
T4	33	12 (36.3%)	18 (54.5%)	11 (33.3%)	7 (21.2%)	6 (18.1%)	12 (36.3%)
		*p* = 0.001	*p* = 0.001	*p* = 0.001	*p* = 0.001	*p* = 0.001	*p* = 0.001
Depth							
Mucosa	20	0	0	0	0	0	0
Submucosa	12	0	0	0	0	0	0
Muscolaris propria	44	2 (4.5%)	5 (11.3%)	2 (4.5%)	2 (4.5%)	0	2 (4.5%)
Subserosa	53	11 (20.7%)	23 (43.3%)	10 (18.8%)	15 (28.3%)	11 (20.7%)	16 (30.1%)
Serosa	33	12 (36.3%)	18 (54.5%)	11 (33.3%)	7 (21.2%)	6 (18.1%)	12 (36.3%)
		*p* = 0.001	*p* = 0.002	*p* = 0.001	*p* = 0.001	*p* = 0.001	*p* = 0.01
N status							
N1	56	0	0	0	0	0	0
N2	48	2 (4.1%)	7 (14.5%)	0	5 (10.4%)	1 (2.0%)	0
N3a	44	13 (29.5%)	28 (63.6%)	10 (22.7%)	12 (27.2%)	9 (20.4%)	23 (52.7%)
N3b	14	10 (71.4%)	11 (78.5%)	13 (92.8%)	7 (50.0%)	7 (50.0%)	7 (50.0%)
		*p* = 0.001	*p* = 0.001	*p* = 0.001	*p* = 0.001	*p* = 0.01	*p* = 0.001
AJCC/TNM stage							
IB	32	0	0	0	0	0	0
IIA	16	0	0	0	0	0	0
IIB	29	2 (6.8%)	2 (6.8%)	0	2 (6.9%)	0	0
IIIA	30	0	8 (26.6%)	2 (6.6%)	3 (10.0%)	1 (3.3%)	2 (6.6%)
IIIB	41	13 (31.7%)	25 (60.9%)	8 (19.5%)	12 (29.2%)	9 (21.9%)	21 (51.2%)
IIIC	14	10 (71.4%)	11 (78.5%)	13 (92.8%)	7 (50.0%)	7 (50.0%)	7 (50.0%)
		*p* = 0.001	*p* = 0.005	*p* = 0.001	*p* = 0.001	*p* = 0.01	*p* = 0.05

Values are mean ± SD. *p* values: comparison with values in overall patients. All the patients were included in all evaluations. Numbers in brackets represents percentage. LN: lymph node. M: middle third. L: lower third. Gre: greater curvature. Less: lesser curvature. NA: not applicable.

**Table 7 cancers-15-02139-t007:** Multivariate analysis.

Parameters	OR	(95% CI)	*p*
Age	0.850	(0.342–2.112)	0.7
Sex	1.207	(0.491–2.966)	0.6
Tumor size	0.874	(0.290–2.634)	0.8
Lauren	3.067	(1.180–7.971)	0.02
Tumor grading	0.768	(0.291–2.025)	0.5
Circular location	3.279	(1.222–8.797)	0.01
Longitudinal location	0.170	(0.059–0.489)	0.001
T stage	0.109	(0.034–0.497)	0.001
Depth	0.107	(0.028–0.399)	0.001

OR: Odds Ratio; CI: confidence interval.

## Data Availability

The authors confirm that the data supporting the findings of this study are available within the article.

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
