# Peer review of "Pattern of Distribution of Lymph Node Metastases in Individual Stations in Middle and Lower Gastric Carcinoma"

_cancers, 2023, doi:10.3390/cancers15072139_

Round 1
Reviewer 1 Report
I thank the authors for the opportunity to review their manuscript entitled "Pattern of distribution of lymph node metastases in individual stations in distal gastric carcinoma."
The premise of this works relies on improving our understanding of various lymphatics drainage patterns in cancers of the mid and distal stomach as means for potential tailoring or customizing the lymphadenectomy.
The main limitation are that this topic has been published upon previously:
Khalayleh H, Kim Y, Man Yoon H, Ryu KW, Kook M. Evaluation of Lymph Node Metastasis Among Adults With Gastric Adenocarcinoma Managed With Total Gastrectomy. JAMA Netw Open. 2021;4(2):e2035810.
Choi YY, An JY, Katai H, Seto Y, Fukagawa T, Okumura Y, Kim DW, Kim HI, Cheong JH, Hyung WJ, Noh SH. A Lymph Node Staging System for Gastric Cancer: A Hybrid Type Based on Topographic and Numeric Systems. PLoS One. 2016 Mar 11;11(3):e0149555.
Lu, H., Zhao, B., Huang, R. et al. Central lymph node metastasis is predictive of survival in advanced gastric cancer patients treated with D2 lymphadenectomy. BMC Gastroenterol 21, 15 (2021).
--------------------------------------------------
Overall, the paper is clear and well-written. The data are presented in a clear manner.
Some of the inherent limitations of this study include its retrospective design.
In the conclusions, the authors summarize that "The knowledge of the localization of the tumor could lead to a tailored lymphadenectomy in the case of small tumors, in consideration of the absence of involvement of stations 7 to 12 in distal T1 cancers". However, two main things interfere with this claim. (1) the retrospective nature of this work predisposes potential biases and such determinations are better suited in a prospective study. (2) T1 staging was determined postoperatively - Limiting the applicability of pre/intra operative decision making.
Author Response
My co-authors and I thank the reviewer for their appreciation of our study.
We are aware that our study has limitations. Firstly, the sample size is small. Furthermore, the retrospective nature of this study has its potential source of intrinsic biases.
Our results must be interpreted with caution because it represents only a group of patients with distal gastric carcinoma. We excluded from the evaluation cases with adenocarcinoma of the upper-third of the stomach (Siewert III cancer).
We are aware that it is commonly accepted that these tumors should be treated in the same way as middle-third or lower-third gastric carcinoma (total gastrectomy with removal of the distal esophagus at least 5 cm above the macroscopic extent of the cancer) but were not included in the present study because the lymphatic flow in these is directed towards groups of lymph nodes excluded from the standard D2 dissection. However, seeing the limited research regarding a more tailored approach in the treatment of distal cancer, this study provides promising results and demonstrates the necessity for further research concerning the extent of lymphadenectomy. This preliminary work will need to be followed up with prospective and multicenter studies.
My authors and I agree that most T1 tumors are diagnosed postoperatively. However, we cannot forget that in many high-volume centres, with new diagnostic methods, it is also possible to obtain a preoperative diagnosis of T1.
We are aware that other studies in the literature, including on larger series, have examined the distribution of lymph node metastases in the individual stations. The cited works were evaluated by us. Our work does not focus on the evaluation of patients undergoing total gastrectomy only, nor does it analyze only metastases at the level of central lymph node stations.
Reviewer 2 Report
This exciting study explores the distribution pattern of positive lymph nodes after radical gastrectomy for distal gastric carcinoma. Furthermore, potential predictors of positive lymph node distribution were explored. Although there is no novelty to the field with this study, the paper would be of potential interest to the readers because there are few studies addressing the same topic with Western patients.
Major concerns:
There is a relatively small number of patients during a pretty long period.
The title refers to distal gastric carcinoma. However, the study also includes patients with mid-portion tumors. The title should be changed to reflect the study better.
The simple summary appears to be quite different from the Abstract. Please consider putting it in accordance.
There is no new data from his study, and it is not clear how to use the result of the present study for clinical decision-making
Author Response
My co-authors and I thank the reviewer for their appreciation of our study.
We are aware that our study has limitations. Firstly, the sample size is small. Furthermore, the retrospective nature of this study has its potential source of intrinsic biases. Our results must be interpreted with caution because it represents only a group of patients with distal gastric carcinoma. We excluded from the evaluation cases with adenocarcinoma of the upper-third of the stomach (Siewert III cancer). We are aware that it is commonly accepted that these tumors should be treated in the same way as middle-third or lower-third gastric carcinoma (total gastrectomy with removal of the distal esophagus at least 5 cm above the macroscopic extent of the cancer) but were not included in the present study because the lymphatic flow in these is directed towards groups of lymph nodes excluded from the standard D2 dissection. However, seeing the limited research regarding a more tailored approach in the treatment of distal cancer, this study provides promising results and demonstrates the necessity for further research concerning the extent of lymphadenectomy. This preliminary work will need to be followed up with prospective and multicenter studies.
We are aware that other studies in the literature, including on larger series, have examined the distribution of lymph node metastases in the individual stations. The cited works were evaluated by us. Our work does not focus on the evaluation of patients undergoing total gastrectomy only, nor does it analyze only metastases at the level of central lymph node stations.
We are aware that there are few Western case studies that have evaluated this aspect.
The title has been changed and the changes have been highlighted in yellow.
The summary has been modified and the changes have been highlighted in yellow.
Reviewer 3 Report
The authors present an interesting statistical study on the distribution of lymph node metastases in distal gastric carcinoma. It is a very comprehensive study, aiming to determine the influence of multiple factors (age, tumor size, grading, location, the depth of invasion) on metastatic lymph node distribution.
Even it is statistical study, it can help to guide the surgical treatment through a tailored lymphadenectomy.
My suggestion is like, for a more easier observation of lymph node metastatis per nodal station (tables 5 and 6), an analysis through K-means clustering to be included, at least for the most important factors.
Considering the above my recommendation is for Minor Revisions.
Author Response
My co-authors and I thank the reviewer for their appreciation of our study.
We have not included further statistical analyzes in tables 5 and 6 to avoid making the tables themselves unintelligible.
We are aware that our study has limitations. Firstly, the sample size is small. Furthermore, the retrospective nature of this study has its potential source of intrinsic biases.
Our results must be interpreted with caution because it represents only a group of patients with distal gastric carcinoma. We excluded from the evaluation cases with adenocarcinoma of the upper-third of the stomach (Siewert III cancer).
We are aware that it is commonly accepted that these tumors should be treated in the same way as middle-third or lower-third gastric carcinoma (total gastrectomy with removal of the distal esophagus at least 5 cm above the macroscopic extent of the cancer) but were not included in the present study because the lymphatic flow in these is directed towards groups of lymph nodes excluded from the standard D2 dissection.
However, seeing the limited research regarding a more tailored approach in the treatment of distal cancer, this study provides promising results and demonstrates the necessity for further research concerning the extent of lymphadenectomy. This preliminary work will need to be followed up with prospective and multicenter studies.
Round 2
Reviewer 2 Report
The authors correctly addressed all significant concerns raised by the reviewers. Although there is no novelty to the field with this study, a study including Western patients deserves to be considered for publication. Thus, I would reconsider my initial decision.